# Outcomes and Complications from a Randomized Controlled Study Comparing Conventional Stent Placement Versus No Stent Placement after Ureteroscopy for Distal Ureteric Calculus < 1 cm

**DOI:** 10.3390/jcm11237023

**Published:** 2022-11-28

**Authors:** Suraj Jayadeva Reddy, Bathi Sourabh Reddy, Arun Chawla, Jean J. M. C. H. de la Rosette, Pilar Laguna, Padmaraj Hegde, Abhijit Shah, Anupam Choudhary, Sanket Kankaria, Vivekanand Kedarlingayya Hiremath

**Affiliations:** 1Department of Urology and Renal Transplantation, Kasturba Medical College, MAHE, Manipal 576104, Karnataka, India; 2Istanbul Medipol Mega University Hospital, Istanbul 34214, Turkey

**Keywords:** conventional stent, stent-related symptoms, quality of life, LUTS, sexual function

## Abstract

Ureteric stent insertion following ureteroscopic lithotripsy (URSL) is a common and widely accepted procedure. However, there is no agreement on whether a ureteric stent should be placed following an uncomplicated URSL. Furthermore, the definition of uncomplicated URSL remains debatable. To compare the efficacy, safety, and morbidity of no stent placement with the conventional stent placement after uncomplicated retrograde semirigid URS for a distal ureteric calculus of size ≤1 cm, we compared the corresponding complication rates, emergency visits, secondary interventions, and pain at follow-up. Following an uncomplicated ureteroscopic lithotripsy, 104 patients were randomized into the conventional stented group (CSG) and nonstented group (NSG). Lower urinary tract symptoms and sexual function were evaluated using validated questionnaires (IPSS + IIEF-5 + MSHQ-EjD/FSFI) preoperatively and at 4 weeks during follow-up. Pain scores at follow-up were recorded using a visual analogue scale (VAS). Patients who visited the emergency room or needed secondary interventions before the recommended follow-up time were noted. The Generalized Estimating Equations method was used to explore the difference in change in the domains of IPSS, IIEF-5, MSHQ-EjD, and FSFI between the two groups over time. A significant difference was noted in the following IPSS domains: Frequency, Urgency, Nocturia, Storage Symptoms, Total IPSS Score (*p* ≤ 0.001), and QoL (*p* = 0.002); IIEF-5 domains: Overall Score (*p* = 0.004); MSHQ-EjD domains: Ejaculation Bother/Satisfaction (*p* ≤ 0.001); and FSFI domains: Lubrication (*p* ≤ 0.001), Satisfaction (*p* = 0.006), and Overall Score (*p* = 0.004). There was no significant difference between the various groups in terms of distribution of emergency visits, readmission and secondary interventions, pain at follow-up (VAS), and need for long-term analgesia. Nonplacement of stents after uncomplicated URS decreases stent-related symptoms and preserves QoL without placing the patient under increased postoperative risk.

## 1. Introduction

Urolithiasis poses a clinical and economic burden on healthcare systems, and it is frequently associated with a high rate of recurrence and a significant impact on quality of life [1,2]. Ureteric stent insertion following ureteroscopic lithotripsy (URSL) is a common and widely accepted procedure [3]. In complicated URS involving bleeding, ureteric trauma, or a large residual stone burden, ureteric stenting is recommended [4]. However, there is no agreement on whether a ureteric stent should be placed routinely following a simple ureteroscopy for stone retrieval, whereas the definition of uncomplicated URSL is still debatable [5]. Despite this controversy, most urologists routinely insert ureteric stents, justified by the hypothetical fact that stent placement promotes the passage of residual stone fragments and clots, presumably lowers the risk of stricture formation, and prevents ureteric obstruction and renal colic induced by ureteric oedema [6,7]. Nonetheless, conventional double-J (DJ) stent insertion is associated with flank pain during voiding, infection, and irritative voiding symptoms, as well as a decline in patient quality of life (QoL) due to stent-related symptoms (SRS) [8]. DJ stents have been reported in various studies to induce bladder discomfort due to the intravesical loop [9].

Therefore, we hypothesized that avoiding a routine conventional stent placement following an uncomplicated URSL for distal ureteric calculus is feasible, thereby avoiding SRS and other stent-related complications. Our aim is to compare the efficacy, safety, and morbidity of no stent placement with the conventional stent placement after uncomplicated retrograde semi rigid URSL for a distal ureteric calculus of size ≤ 1 cm. Additionally, we compare the stent-related complication rate, emergency visits, secondary interventions, and pain at the time of follow-up using a visual analogue scale (VAS) between the two groups.

## 2. Materials and Methods

### 2.1. Study Design and Ethics Statement

This was a prospective, single-center, randomized study conducted from September 2020 to March 2022. Institutional ethics committee (IEC: 225/2020) approval and CTRI approval (CTRI/2020/09/027530) were obtained prior to initiation of the study. The sample size was calculated using PASS software, with a power of 80%, a significance level of 0.05, and level of confidence of 95%. In total, 104 consecutive patients were randomized into two groups (group 1, CSG; group 2, NSG) in a ratio of 1:1 using a computer-generated simple randomization method. The randomization sequence was concealed using the sequentially numbered, opaque sealed envelopes (SNOSE) method. Randomization was performed intraoperatively (after lithotripsy but before stent placement) and was revealed to the operating surgeon.

A brief overview of the methodology is presented in Figure 1.

### 2.2. Study Participants

Consecutive patients > 18 years of age with symptomatic unilateral distal ureteral stones ≤ 10 mm in diameter who underwent ‘Uncomplicated URSL’ were included in the study. Patients in any of the following groups were excluded from the study: (1) pediatric age group/pregnancy or breast feeding; (2) patients undergoing bilateral procedures; (3) patients who had a pre-existing ureteral stent at the time of ureteroscopy; (4) patients with anatomical abnormalities (horseshoe kidney, pelvic kidney, solitary kidney, etc.) and transplant patients; (5) patients with systemic signs of infection (sepsis); (6) patients from whom there was difficulty in obtaining consent; (7) procedures complicated by submucosal passage, perforation of the ureter, or gross bleeding (PULS Grade > 2); (8) patients in whom stone fragments were not completely retrieved during primary URSL.

### 2.3. Intervention

Under spinal or general anesthesia, the patient was placed in the dorsal lithotomy position. Cystoscopy was performed using a 30° scope (Storz, Tuttlingen, Germany) and sheath of size 20/22 Fr. Methodical urethrocystoscopy was performed, and bilateral ureteric orifices were visualized. A Teflon-coated guidewire with a hydrophilic tip (0.032 inch, Blueneem, Harohalli, India) was then passed into the ureter beyond or up to the calculus. The position was confirmed on fluoroscopy. After removing the cystoscope, an infant feeding tube of 8 Fr was inserted. Then, a 6/7.5 Fr semirigid ureteroscope (Richard Wolf, Knittlingen, Germany) was inserted along the guidewire and negotiated up to the ureteric calculus. Patients with edema of the ureteric orifice, inflammation, or a narrow orifice requiring balloon dilatation before negotiating were not included in the study. After proper visualization, the stone was fragmented using a pneumatic lithotripter (Swiss Lithoclast Master, Hyderabad, India). The stone fragments were washed out completely by gravity irrigation or removed with graspers. The ureteric mucosal appearance post procedure was graded using the Post Ureteroscopic Lesion Scale (PULS) endoscopic visual grading system. A PULS grading of >2 was considered complicated and excluded from the study. The absence of residual fragments was confirmed by the operating surgeon endoscopically and fluoroscopically, and those patients in whom residual fragments were present were excluded from the study. Patients who underwent ureteroscopy and lithotripsy for uncomplicated ureteric calculi were stratified into two groups. Among the total of 104 patients, 52 patients underwent conventional DJ stent placement, while in the remaining 52 patients, no DJ stent was placed. The stent size used was either 6 Fr/5 Fr, while the length of 26 cm or 24 cm was decided by the operating surgeon based on ureteral caliber and the distance from UPJ to VUJ or the height of the patient. Patients were discharged within one postoperative day with oral antibiotics and analgesics. The standard analgesic used in our study was oral tramadol on demand.

### 2.4. Postoperative Follow-Up

Four weeks after the primary procedure, patients of the CSG group were advised to come for stent removal, and those of the NSG group were asked to come for follow-up. During this follow-up, patients were asked to fill out questionnaires pertaining to their postoperative status. Patients who visited the emergency room and/or needed secondary interventions before the advised follow-up time were noted. Imaging by X-ray (for radio-opaque stones) and ultrasound (US)/NCCT (for radiolucent stones) was performed to check for any significant residual fragments (≥3 mm) and the need for a repeat or ancillary procedure. Patients requiring repeat procedures were noted in both groups.

### 2.5. Statistical Analysis

Statistical analysis was performed using IBM SPSS Statistics 25. The chi-square test was used to check for significant effects between the groups to obtain a comparison. Descriptive statistics were obtained for the mean score, standard deviation, and confidence interval. To indicate statistical significance, a *p* value of <0.05 was taken. The Generalized Estimating Equations method was used to explore the difference in change in the domains of IPSS, IIEF-5, MSHQ-EjD, and FSFI between the two groups over time.

## 3. Results

The CONSORT study principles were followed, as shown in Figure 2 [10]. Of the 136 consecutive patients with distal ureteric calculus, 104 patients met the inclusion criteria and were randomized into two study groups. Three patients were excluded from final analysis from the CSG group. All three patients had their stent removed at another hospital and were unavailable to fill out the questionnaires post operatively. Nine patients were excluded from the NSG group as they had only a telephonic follow up during the lockdown period. A total of 92 patients remained for analysis, including 49 in the CSG group and 43 in the NSG group. All patient variables (age, gender) and stone parameters (size, laterality, location) were similar in both groups, with no statistically significant difference noted (*p* > 0.05). Among the 92 patients in our study, 30 (32.6%) had a score of 0, 60 (65.2%) had a score of 1, and 2 (2.2%) had a score of 2 on the Post Ureteroscopic Lesion Scale (PULS) (Appendix A).

### 3.1. Lower Urinary Tract Symptom Domains

The variable domains of IPSS at baseline and during follow-up were not normally distributed in the two subgroups of the variable Group. Thus, nonparametric tests (Wilcoxon–Mann–Whitney U Test) were used to make group comparisons. There was no significant difference between the two study groups (CSG and NSG) at baseline for all the domains of IPSS. Also, at follow-up, there was no significant difference between the groups in the following IPSS domains: Incomplete Emptying (W = 1059.000, *p* = 0.954), Intermittency (W = 1200.500, *p* = 0.151), Weak Stream (W = 1143.000, *p* = 0.300), Straining (W = 1032.000, *p* = 0.840), and Voiding Symptoms (W = 1167.000, *p* = 0.327). However, there was a significant difference between the two groups (CSG and NSG) in terms of Frequency (W = 1516.000, *p* = 0.001), Urgency (W = 1350.000, *p* = 0.015), Nocturia (W = 1545.000, *p* = 0.001), Storage Symptoms (W = 1538.500, *p* = 0.001), Total (W = 1427.000, *p* = 0.003), and QoL Score (W = 1297.500, *p* = 0.047), with the median being highest in the CSG. The Wilcoxon signed-rank test was used to explore the difference in various domains of IPSS between the time points (at baseline and at follow-up) within each group. The Generalized Estimating Equations method was used to explore the difference in change in the domains of IPSS between the two groups over time. Significant differences were noted in the following IPSS domains: Frequency (*p* ≤ 0.001), Urgency (*p* ≤ 0.001), Nocturia (*p* ≤ 0.001), Storage Symptoms (*p* ≤ 0.001), Total IPSS Score (*p* ≤ 0.001), and QoL (*p* = 0.002) (Table 1 and Appendix A). 

### 3.2. Sexual Function Domains

Sexual function evaluations were performed in men (n = 69) using IIEF-5 and MSHQ-EjD and in women (n = 23) using the FSFI questionnaire. The variable domains of IIEF-5, MSHQ-EjD, and FSFI were not normally distributed in the two subgroups of the variable Group. Thus, nonparametric tests (Wilcoxon–Mann–Whitney U Test) were used to make group comparisons. There was no significant difference between the two study groups at baseline for all the domains of IIEF-5, MSHQ-EjD, and FSFI.

The Wilcoxon–Mann–Whitney test was used to compare the two groups (CSG and NSG) at each of the time points (at baseline and at follow-up). The Generalized Estimating Equations method was used to explore the difference in change between the two groups over time in the IIEF-5 domains Maintenance Frequency (*p* = 0.030), Maintenance Ability (*p* = 0.035), and Overall Score (*p* = 0.004); the MSHQ-EjD domains Ejaculatory Function Score (*p* = 0.010) and Ejaculation Bother/Satisfaction (*p* = <0.001); and the FSFI domains Desire (*p* = 0.012), Arousal (*p* = 0.038), Lubrication (*p* = <0.001), Satisfaction (*p* = 0.006), and Overall Score (*p* = 0.004) (Table 2 and Table 3).

### 3.3. Postoperative Pain and Complications

The assessment of a given intervention’s safety profile necessitates patient follow-up regarding the most common complications that are likely to occur. Table 4 summarizes the complications seen in both groups.

The chi-squared test was used to explore the association between ‘Groups’ and ‘Complications’. There was no significant difference between the various groups in terms of the distribution of emergency visits (χ^2^ = 0.143, *p* = 0.706), readmission (χ^2^ = 1.519, *p* = 0.367), and secondary interventions (conservative management (χ^2^ = 1.152, *p* = 0.467), early stent removal (χ^2^ = -, *p* = -), relook URS (χ^2^ = 0.887, *p* = 1.000), and PCN insertion). It was also noted there was no significant difference between the various groups in terms of the distribution of analgesic requirement of >5 days (χ^2^ = 1.867, *p* = 0.172) and pain at follow-up (VAS) (W = 1271.500, *p* = 0.066).

## 4. Discussion

In this study, we aimed to evaluate whether the routine placing of a stent could be safely omitted in patients who had an uncomplicated URSL for distal ureteral stones of size up to 1.0 cm. We confirmed significantly less postoperative discomfort in the nonstented group with no increased risk of complications at short-term follow-up. It is important to be aware that we used a strict definition for an uncomplicated URSL. Our study also demonstrated similar postoperative pain complaints and comparable rates of emergency room visits and readmissions between the two groups.

Various hypotheses have been proposed to explain the origin of stent-related symptoms, including local irritation of the neuron-rich trigone by the intravesical portion of the stent, resulting in smooth muscle spasms, and SRS could be aggravated by the movement of the distal end of the stent with changes in posture, resulting in local tissue irritation, but none has been successful in precisely defining the cause [11,12]. We used the IPSS questionnaire to assess lower urinary tract symptoms. Patients in the CSG group exhibited a statistically significant worsening of the following IPSS domains: Frequency, Urgency, Nocturia, Storage Symptoms, Total IPSS Score (*p* < 0.001), and QoL (*p* < 0.002). Our findings are consistent with those of earlier research where nonstented patients had better outcomes in terms of urgency, frequency, suprapubic pain, hematuria, postoperative discomfort, and analgesic use [13,14]. Moreover, a meta-analysis of nine randomized controlled trials of ureteroscopy stenting was presented by Nabi et al., in which individuals who were stented had a higher rate of UTIs [15].

Several studies have shown comparable outcomes between unstented and stented groups in terms of early postoperative problems, including low-grade fever, hematuria, urinary tract infection, and flank pain [13,16]. A recent paper reported that stent placement during URS was independently associated with a 25% increase in emergency room visits within 30 days, with flank pain being the most common reason for both groups, but there was no increased risk of hospitalization [17].

On the opposite end of the spectrum, researchers have already attempted to find a strategy to avoid SRS without endangering surgical outcomes. The prospect of lowering SRS by altering stent designs, such as their size, material, softness, placement, and loop completeness, has been discussed, although the evidence is still equivocal [18,19].

Although a few studies which compared conventional DJ stents to complete intraureteral stents have demonstrated a reduction in SRS, enhancement of QoL, and less stent-related discomfort [20,21,22], a similar study that was conducted at our institute found a reduction in symptom scores across LUTS domains of the USSQ, but the reduction was not statistically significant [23].

In addition to lower urinary tract symptoms, the placement of stents is thought to have an influence on sexual function in both men and women, consequently lowering QoL. However, the exact mechanism via which the DJ stent may interfere with sexual function is unknown.

Previously conducted studies which eliminated the routine use of stents following an uncomplicated URSL demonstrated severe impairment of sexual function in both genders, particularly in males, with reduced IIEF scores; however, this was temporary [14]. Among women who were stented, individual domains of the FSFI—the Arousal, Orgasm, and Satisfaction subdomains—were statistically lower in the stented group [24,25,26]. Meanwhile, studies involving complete intraureteral stents with extraction strings by Kim et al. [27] and Shah M et al. [23] demonstrated a general drop in sexual activity among patients following placement of a stent with extraction strings due to stent-related pain, anxiety of stent dislodgement, or a thread dangling from the urethral meatus.

Our study results are in concordance with those of previous studies in that men in the CSG group performed worse in terms of overall IIEF-5 scores (*p* = 0.004) than men in the NSG group. The Maintenance Frequency and Maintenance Ability subdomains presented a statistically significant decrease in scores. Ejaculatory dysfunction was also observed in men, as evidenced by a statistically significant decrease in the scores of two MSHQ domains. Women in the CSG group had poorer outcomes in terms of sexual function, as evidenced by decreases in FSFI scores in the domains of Arousal, Lubrication, Satisfaction, and Overall Scores (*p* = 0.004). We assume that this sexual dysfunction in both men and women is predominantly due to the associated SRS, stent-related pain, and patient anxiety because they assume that they are still experiencing a therapy procedure until the stent is removed.

Finally, office-based cystoscopy is required for the removal of conventional DJ stents [28]. Our study did not include the costs involved with follow-up stent removal, but it merits consideration. Stent insertion after URSL results in extended operative findings, the need for a second procedure to remove the stent, and an increase in the overall costs of surgery [14,29]. Further, a forgotten stent inside the body is associated with negative consequences that can occur in the form of calculus formation, infections, and a requirement of multiple procedures [30]. Moreover, in the present ongoing pandemic due to COVID-19, nonemergencies such as stent removals may have to be delayed, leading to further deterioration of QoL [31]. Such a scenario can be avoided if stent placement could be omitted.

The authors would like to highlight that among the 136 patients screened for the procedure, only 32 (23.52%) were excluded from the study, while the rest of them met the criteria of “uncomplicated” URS. Consequently, the majority of the patients treated for distal ureteral stones can safely avoid the placement of a stent. The authors recommend the use of the included algorithm to decide on the placement of a stent following an “uncomplicated” URS (Figure 3).

## 5. Limitations

Our study had certain limitations. As it was a single-blinded study, the possibility of observation bias cannot be excluded. Only a well-selected cohort of patients undergoing uncomplicated URSL was included in the study. Patients in the stented arm had their stents for 4 weeks, which is quite long and may add to the morbidity. Other domains such as cost analysis and effect of second procedures, including cystoscopy and stent removal, were not assessed. We have also no long-term follow-up data to confirm the claim that there is no increased risk of ureteral stricture or upper tract impairment.

## 6. Conclusions

The unstented approach had the smoothest postoperative course, avoiding the unpleasant side effects of stent-related symptoms, sexual dysfunction in both genders, and the subsequent decline in QoL. We can safely conclude that stent placement is unwarranted in uncomplicated procedures with no increased risk of postoperative sequelae. With the omission of stent placement and a second procedure to retrieve it, the procedure is also more cost-effective. Further studies are required to prove the long-term effects and to see whether reducing the duration of the stents improves outcomes.

## Figures and Tables

**Figure 1 jcm-11-07023-f001:**
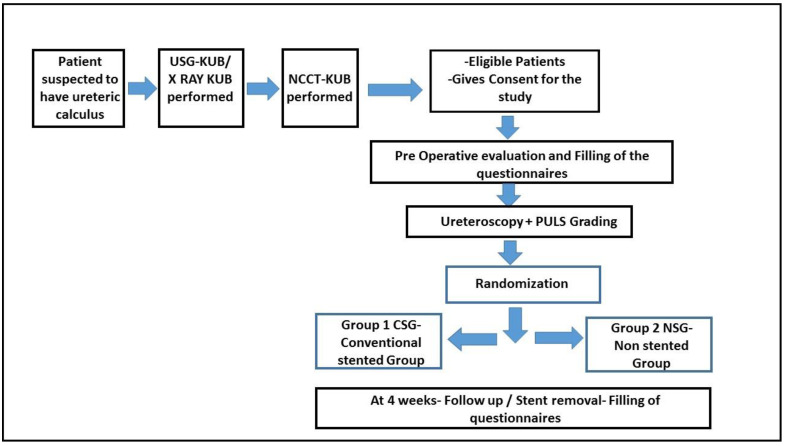
Methodology of the study.

**Figure 2 jcm-11-07023-f002:**
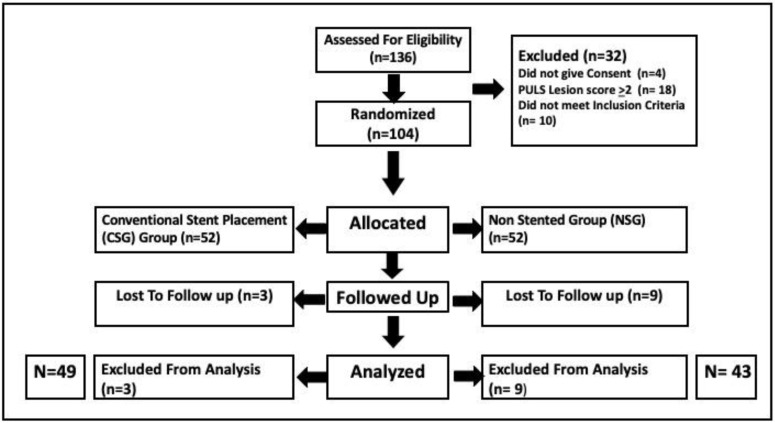
CONSORT diagram of our study.

**Figure 3 jcm-11-07023-f003:**
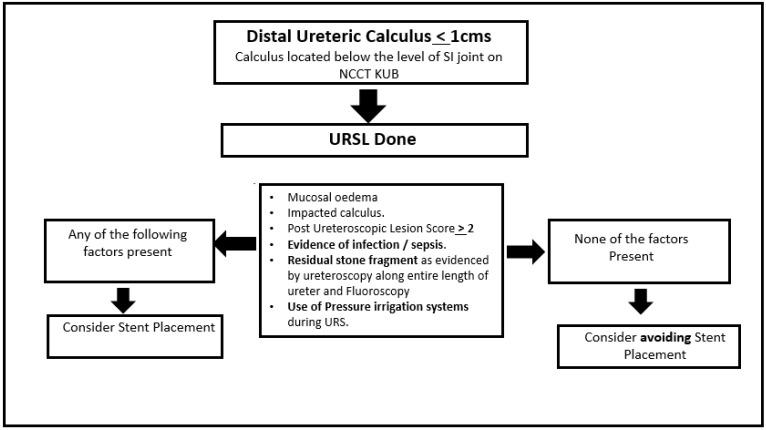
Algorithm proposed regarding the placement of stents after uncomplicated URS.

**Table 1 jcm-11-07023-t001:** Comparison of the two groups in terms of change in IPSS (n = 92).

		Group		*p* Value for Comparison of the Two Groups at Each of the Time Points (Wilcoxon–Mann–Whitney Test)	Overall *p* Value for Comparison of Change (Generalized Estimating Equations)
**IPSS: Incomplete Emptying**		**CSG-Mean (SD)**	**NSG-Mean (SD)**		**0.716**
	Baseline	0.20 (0.50)	0.19 (0.39)	0.854	
	Follow-Up	0.22 (0.51)	0.19 (0.39)	0.954	
	***p* Value for change over time within each group (Wilcoxon Test)**	0.766	1.000		
**IPSS:** **Frequency**		**NSG-Mean (SD)**	**CSG-Mean (SD)**		**<0.001**
	Baseline	0.65 (0.75)	0.56 (0.77)	0.472	
	Follow-Up	1.20 (0.91)	0.51 (0.83)	<0.001	
	***p* Value for change over time within each group (Wilcoxon Test)**	**<0.001**	0.588		
**IPSS: Intermittency**		**NSG-Mean (SD)**	**CSG-Mean (SD)**		**0.058**
	Baseline	0.24 (0.43)	0.26 (0.49)	0.945	
	Follow-Up	0.39 (0.53)	0.23 (0.43)	0.151	
	***p* Value for change over time within each group (Wilcoxon Test)**	0.057	1.000		
**IPSS: Urgency**		**NSG-Mean (SD)**	**CSG-Mean (SD)**		**<0.001**
	Baseline	0.67 (0.75)	0.74 (0.69)	0.551	
	Follow-Up	1.33 (1.03)	0.84 (0.87)	0.015	
	***p* Value for change over time within each group (Wilcoxon Test)**	**<0.001**	0.203		
**IPSS: Weak Stream**		**NSG-Mean (SD)**	**CSG-Mean (SD)**		**0.108**
	Baseline	0.12 (0.33)	0.12 (0.32)	0.933	
	Follow-Up	0.22 (0.42)	0.14 (0.35)	0.300	
	***p* Value for change over time within each group (Wilcoxon Test)**	**0.037**	1.000		
**IPSS: Straining**		**NSG-Mean (SD)**	**CSG-Mean (SD)**		**0.198**
	Baseline	0.18 (0.39)	0.33 (0.52)	0.173	
	Follow-Up	0.35 (0.56)	0.37 (0.58)	0.840	
	***p* Value for change over time within each group (Wilcoxon Test)**	0.052	1.000		
**IPSS: Nocturia**		**NSG-Mean (SD)**	**CSG-Mean (SD)**		**<0.001**
	Baseline	1.10 (0.68)	1.07 (0.67)	0.795	
	Follow-Up	1.71 (0.76)	1.00 (0.72)	<0.001	
	***p* Value for change over time within each group (Wilcoxon Test)**	**<0.001**	0.299		
**IPSS: Voiding Symptoms**		**NSG-Mean (SD)**	**CSG-Mean (SD)**		**0.135**
	Baseline	0.78 (1.34)	0.88 (1.45)	0.942	
	Follow-Up	1.18 (1.60)	0.95 (1.53)	0.327	
	***p* Value for change over time within each group (Wilcoxon Test)**	0.039	0.789		
**IPSS: Storage Symptoms**		**NSG-Mean (SD)**	**CSG-Mean (SD)**		**<0.001**
	Baseline	2.47 (2.03)	2.37 (1.84)	0.965	
	Follow-Up	4.22 (2.38)	2.33 (2.03)	<0.001	
	***p* Value for change over time within each group (Wilcoxon Test)**	**<0.001**	0.416		
**IPSS: Total Score**		**NSG-Mean (SD)**	**CSG-Mean (SD)**		**<0.001**
	Baseline	3.24 (3.17)	3.26 (3.02)	0.855	
	Follow-Up	5.37 (3.83)	3.23 (3.30)	0.003	
	***p* Value for change over time within each group (Wilcoxon Test)**	**<0.001**	0.394		
**IPSS: QoL Score**		**NSG-Mean (SD)**	**CSG-Mean (SD)**		**0.002**
	Baseline	0.67 (0.90)	0.95 (1.31)	0.575	
	Follow-Up	1.59 (1.51)	1.07 (1.42)	0.047	
	***p* Value for change over time within each group (Wilcoxon Test)**	**<0.001**	0.374		

**Table 2 jcm-11-07023-t002:** Comparison of the two groups in terms of change in M-IIEF-5 and MSHQ-EjD (n = 69).

		Group		*p* Value for Comparison of the Two Groups at Each of the Time Points (Wilcoxon–Mann–Whitney Test)	Overall *p* Value for Comparison of Change in M-IIEF-5: between the Two Groups (Gen. Estimating Equations)
		CSG-Mean (SD)	NSG-Mean (SD)		
**IIEF-5:** **Erection Confidence**	**Baseline**	**4.72 (0.57)**	4.55 (0.75)	0.300	**0.091**
	Follow-Up	4.61 (0.60)	4.58 (0.79)	0.772	
	** *p* ** ** Value for change over time within each group (Wilcoxon Test)**	0.129	0.773		
**IIEF-5: Erection Firmness**	Baseline	4.61 (0.64)	4.52 (0.67)	0.483	**0.098**
	Follow-Up	4.36 (0.68)	4.42 (0.83)	0.458	
	** *p* ** ** Value for change over time within each group (Wilcoxon Test)**	0.008	0.149		
**IIEF-5: Maintenance Frequency**	Baseline	4.58 (0.65)	4.45 (0.75)	0.466	**0.030**
	Follow-Up	4.39 (0.73)	4.48 (0.76)	0.484	
	** *p* ** ** Value for change over time within each group (Wilcoxon Test)**	0.011	0.777		
**IIEF-5:** **Maintenance Ability**	Baseline	4.56 (0.61)	4.42 (0.66)	0.400	**0.035**
	Follow-Up	4.31 (0.71)	4.42 (0.79)	0.353	
	** *p* ** ** Value for change: over time within each group (Wilcoxon Test)**	0.008	1.000		
**IIEF-5:** **Intercourse satisfaction**	Baseline	4.44 (0.65)	4.39 (0.66)	0.737	**0.055** **0.004**
	Follow-Up	4.19 (0.62)	4.36 (0.86)	0.119	
	** *p* ** ** Value for change over time within each group (Wilcoxon Test)**	0.008	0.777		
**IIEF-5:** **Overall Score**	Baseline	23.00 (2.63)	22.39 (3.19)	0.408	
	Follow-Up	21.94 (2.83)	22.27 (3.66)	0.316	
	** *p* ** ** Value for change over time within each group (Wilcoxon Test)**	<0.001	0.518		
**MSHQ-EjD: Ejaculatory Function Score**	Baseline	14.22 (1.10)	13.45 (1.46)	0.010	**0.010**
	Follow-Up	13.69 (1.33)	13.42 (1.46)	0.427	
	** *p* ** ** Value for change over time within each group (Wilcoxon Test)**	0.002	0.812		
**MSHQ-EjD: Ejaculation Bother/Satisfaction**	Baseline	0.53 (0.74)	0.76 (0.83)	0.240	**<0.001**
	Follow-Up	1.06 (0.98)	0.76 (0.90)	0.197	
	** *p* ** ** Value for change over time within each group (Wilcoxon Test)**	0.001	1.000		

**Table 3 jcm-11-07023-t003:** Comparison of the two groups in terms of change in FSFI (n = 23).

		Group		*p* Value for Comparison of the Two Groups at Each of the Time Points (Wilcoxon–Mann–Whitney Test)	Overall *p* Value for Comparison of Change in FSFI between the Two Groups (Gen Estimating Equations)
		CSG-Mean (SD)	NSG-Mean (SD)		
**FSFI—Desire**	**Baseline**	**4.92 (0.28)**	5.20 (0.42)	0.080	**0.012**
	Follow-Up	4.69 (0.48)	5.40 (0.52)	0.006	
	** *p* ** ** Value for change over time within each group (Wilcoxon Test)**	0.149	0.346		
**FSFI—Arousal**	Baseline	5.15 (0.69)	4.80 (0.63)	0.225	**0.038**
	Follow-Up	4.46 (0.52)	4.60 (0.52)	0.543	
	** *p* ** ** Value for change over time within each group (Wilcoxon Test)**	0.018	0.346		
**FSFI—Lubrication**	Baseline	5.00 (0.58)	4.80 (0.63)	0.440	**<0.001**
	Follow-Up	4.46 (0.66)	5.00 (0.82)	0.138	
	** *p* ** ** Value for change over time within each group (Wilcoxon Test)**	0.026	0.346		
**FSFI—Orgasm**	Baseline	4.77 (0.44)	4.60 (0.52)	0.414	**0.330**
	Follow-Up	4.46 (0.52)	4.50 (0.71)	0.725	
	** *p* ** ** Value for change over time within each group (Wilcoxon Test)**	0.072	0.773		
**FSFI—Satisfaction**	Baseline	4.85 (0.80)	4.70 (0.48)	0.757	**0.006**
	Follow-Up	4.23 (0.44)	4.70 (0.48)	0.030	
	** *p* ** ** Value for change over time within each group (Wilcoxon Test)**	0.015	1.000		
**FSFI—Pain**	Baseline	5.08 (0.76)	4.90 (0.32)	0.509	**0.443**
	Follow-Up	4.69 (0.63)	4.70 (0.48)	0.914	
	** *p* ** ** Value for change over time within each group (Wilcoxon Test)**	0.120	0.346		
**FSFI—Overall**	Baseline	29.77 (2.83)	29.00 (2.16)	0.616	**0.004**
	Follow-Up	27.00 (2.27)	28.90 (3.00)	0.157	
	** *p* ** ** Value for change over time within each group (Wilcoxon Test)**	0.006	0.533		

**Table 4 jcm-11-07023-t004:** Postoperative complications in CSG and NSG.

Variable		CSG Group (n = 49)	NSG Group (n = 43)	*p*-Value	χ^2^
**Emergency visits**		7 (14.3%)	5 (11.6%)	0.706	0.143
**Readmissions**		4 (8.2%)	1 (2.3%)	0.367	1.519
**Secondary interventions**	**Conservative management**	0 (0.0%)	1 (2.3%)	0.467	1.152
	**Early stent removal**	3 (6.1%)	NA	-	-
	**PCN insertion**	0 (0.0%)	0 (0.0%)	-	-
	**Relook URS**	1 (2.0%)	0 (0.0%)	1.000	0.887
**Analgesia requirement (>5 days)**		11 (22.4%)	5 (11.6%)	0.172	1.867

## Data Availability

The data presented in this study are available on request from the corresponding author. The data are not publicly available due to institutional policy.

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
