# Peer review of "Outcomes and Complications from a Randomized Controlled Study Comparing Conventional Stent Placement Versus No Stent Placement after Ureteroscopy for Distal Ureteric Calculus < 1 cm"

_jcm, 2022, doi:10.3390/jcm11237023_

Round 1

Reviewer 1 Report

Dear authors,

I read with interest the paper entitled "Outcomes and Complications from a Randomized Controlled 2 study comparing Conventional Stent Placement Versus No 3 Stent Placement after Ureteroscopy for Distal Ureteric Calculus 4 < 1 cm"

This is well written and shows interesting answers to a question normally not easily answered.

I have some comments:

How did you define a distal ureteric stone? Are VUJ stones included in those?

How many weeks did patients undergo ureteroscopy after stone diagnosis ?Please specify average waiting time. Have you excluded all cases who had homolateral renal stone or was this not treated? Have you excluded cases of stones found to be impacted once exploring the ureter?

The paper is interesting and the randomisation improved the value of fit. However, all cases were done with a pneumatic device rather than laser. This is considered obsolete and therefore the results will not be as solid as they can be.

You used IPSS and IIEF in you post op questionnaires. Why did you not use the USSQ that is created for stent symptoms? doi:10.1097/01.ju.0000049198.53424.1d.

Regards

Reviewer 2 Report

While the issues of 4-week stent placement and lack of long-term assessment of complications (with imaging postoperatively) are mentioned in the limitations, they should be added to the conclusions as the long-term nature of the stent placement and lack of followup imaging should be factored into the main takeaways from the manuscript. We do not know if the worsened symptoms in the CSG would be lessened if the stents were removed in 1 week. We do not know if all of the NSG patients developed strictures or recurrent issues after 4 weeks postop.

In addition, I'd ask for clarification on the final item of Figure 3 - do you suggest all URS with pressurized irrigation should be stented? It is my practice to use pressurized irrigation for all URS - is that not the case in your institution?

Otherwise, very well done study, just need some editing for English language clarity!
